# Structure of the human lipid-gated cation channel TRPC3

Chen Fan[1†], Wooyoung Choi[1†], Weinan Sun[2‡], Juan Du[1]*, Wei Lü[1]*

[1]Van Andel Institute, Grand Rapids, United States; [2]Vollum Institute, Portland, United States

**Abstract** The TRPC channels are crucially involved in store-operated calcium entry and calcium homeostasis, and they are implicated in human diseases such as neurodegenerative disease, cardiac hypertrophy, and spinocerebellar ataxia. We present a structure of the full-length human TRPC3, a lipid-gated TRPC member, in a lipid-occupied, closed state at 3.3 Angstrom. TRPC3 has four elbow-like membrane reentrant helices prior to the first transmembrane helix. The TRP helix is perpendicular to, and thus disengaged from, the pore-lining S6, suggesting a different gating mechanism from other TRP subfamily channels. The third transmembrane helix S3 is remarkably long, shaping a unique transmembrane domain, and constituting an extracellular domain that may serve as a sensor of external stimuli. We identified two lipid-binding sites, one being sandwiched between the pre-S1 elbow and the S4-S5 linker, and the other being close to the ion-conducting pore, where the conserved LWF motif of the TRPC family is located.
DOI: https://doi.org/10.7554/eLife.36852.001

*For correspondence:
juan.du@vai.org (JD);
wei.lu@vai.org (WL)

[†]These authors contributed equally to this work

Present address: [‡]Janelia Research Campus, Ashburn, United States

Competing interests: The authors declare that no competing interests exist.

## Introduction

The cytosolic-free $Ca^{2+}$ concentration is strictly regulated because calcium is crucial to most cellular processes, from transcription control, to neurotransmitter release, to hormone molecule synthesis (*Berridge et al., 2003*; *Kumar and Thompson, 2011*; *Sudhof, 2012*). A major mechanism regulating calcium homeostasis is store-operated calcium entry (SOCE), which is triggered by the depletion of calcium stored in the endoplasmic reticulum (ER) (*Ong et al., 2016*; *Smyth et al., 2010*). This process activates store-operated channels (SOCs) in the plasma membrane, resulting in the influx of calcium that refills the calcium stores of the ER for further cellular stimulation (*Prakriya and Lewis, 2015*). A key component of SOCE has been identified as the TRPC channels, which are calcium-permeable, nonselective cation channels belonging to the TRP superfamily (*Liu et al., 2003*; *Zhu et al., 1998*; *Zhu et al., 1996*).

Among the seven members in TRPC family, TRPC3, TRPC6, and TRPC7 are the closest homologues, and they are unique in being activated by the lipid secondary messenger diacylglycerol (DAG), a degradation product of the signaling lipid phosphatidylinositol 4,5-bisphosphate (PIP2) (*Itsuki et al., 2012*; *Tang et al., 2001*). However, the molecular mechanism of such activation remains elusive due to a lack of knowledge of the lipid-binding sites. TPRC3, TRPC6, and TRPC7 share several functional domains, including N-terminal ankyrin repeats (AR), a transmembrane domain (TMD) with six transmembrane helixes (S1-S6), and a C-terminal coiled-coil domain (CTD). They also exhibit an unusually long S3 helix, but the function of the S3 helix is poorly understood (*Vazquez et al., 2004*).

TRPC3 is abundantly expressed in the cerebellum, cerebrum, and smooth muscles, and it plays essential roles in the regulation of neurogenesis and extracellular/intracellular calcium signaling (*Gonzalez-Cobos and Trebak, 2010*; *Li et al., 1999*). Dysfunction of TRPC3 has been linked to neurodegenerative disease, cardiac hypertrophy, and ovarian adenocarcinoma (*Becker et al., 2011*; *Kitajima et al., 2016*; *Yang et al., 2009*). Although TRPC3 has wide pharmaceutical applications in

treatment of these diseases, drug development specifically targeting TRPC3 has been limited due to the lack of understanding of its molecular activation mechanisms (*Oda et al., 2017*; *Xia et al., 2015*). Here, we report the structure of full-length human TRPC3 (hTRPC3) in a lipid-occupied, inactive state at an atomic resolution of 3.3 Å using single-particle cryo-electron microscopy (cryo-EM). Our structure revealed the first atomic view of TRPC3 channel and its two lipid-binding sites, providing insight into the mechanisms of lipid activation and regulation of $Ca^{2+}$ homeostasis.

## Results

### Overall architecture

The full-length hTRPC3 could be purified to homogeneity, and electrophysiology experiment showed that the baculovirus-expressed hTRPC3 is functional in HEK293 cells. The three-dimensional reconstruction of hTRPC3 was of sufficient quality to allow de novo modeling of almost the entire protein (*Figure 1*, *Figure 1—figure supplements 1–3*), with the exception of the first 21 N-terminal residues; the region connecting the TRP helix and the C-terminal domain (residues 688–757); the loop connecting the linker domain LD6 and LD7 (residues 281–291); and the last 30 C-terminal residues. We identified two lipid-like densities, one sandwiched between the pre-S1 elbow and the S4-S5 linker, and the other wedged between the P loop and S6 of the adjacent subunit. Notably, we modeled two lipid molecules at these two sites, nevertheless, we were not able to determine the identity of the lipids at current resolution. Interestingly, the TRP helix is perpendicular to the S6, and the density of the hinge region is poorly defined, even though both the TRP helix and S6 exhibit excellent densities (*Figure 1a,b*).

The structure of TRPC3 has a solely alpha-helical composition (*Figure 1a–d*). While TRPC3 shares a similar architecture of the TMD with other TRPCs, the third transmembrane helix, S3, is nearly twice as long as the S3 in any other DAG-insensitive TRPC channels including TRPC1, TRPC4 and TRPC5 (*Figure 1—figure supplement 4*). It elongates into the extracellular space and connects to the S4 through a remarkably long loop, where a glycosylation site is observed (*Figure 1c,d*). The extended S3 gives rise to the shape of TMD distinctive to voltage-gated potassium channels or other TRP channels (*Guo et al., 2017*; *Long et al., 2007*; *Paulsen et al., 2016*; *Shen et al., 2016*; *Winkler et al., 2017*) (*Figure 1e*). Four elbow-like pre-S1 domains extrude from the TMD and are completely buried in detergent micelles, where the lipid 1 density is located (*Figures 1* and *2b*). The C-terminal coiled-coil domain (CTD) is reminiscent of the TRPM4 and TRPM8 structures, having a coiled-coil 'pole' domain in the four-fold symmetry axis, and the 'rib' helix penetrating into a 'tunnel' composed by adjacent intracellular domains (*Figure 1f*). This structure thus stabilizes the tetrameric assembly through hydrophobic and polar interactions (*Figures 1b* and *2b*) (*Winkler et al., 2017*; *Yin et al., 2018*). The ankyrin repeat domain (ARD), located on the bottom of the channel and comprising four pairs of ARs, is significantly smaller than the ARD of TRPA1 and NOMPC (*Jin et al., 2017*; *Paulsen et al., 2016*) (*Figure 2b,c*).

### Transmembrane domain and lipid-binding sites

The TMD of TRPC3 shares topology similar to that of other TRP channels and voltage-gated ion channels, consisting of the S1-S4 domain and the pore domain arranged in a domain-swapped manner (*Figures 1e* and *3a*). Nevertheless, the distinct activation mechanism of TRPC3, TRPC6, and TRPC7 by DAG implies unique features of their TMD. Indeed, comparison of the relative arrangement of the S1-S4 domains with the pore domain shows remarkable differences between TRPC3 and TRPA1 or TRPM4, yet overall agreement with TRPV1 (*Figure 3—figure supplement 1*). Detailed inspection of the TMD in TRPC3 reveals two unique features: a large elbow-like pre-S1 domain harboring a lipid-binding site (lipid 1), and an unusually long S3 helix forming an extracellular domain (ECD), along with the S1-S2 linker and S3-S4 linker (*Figure 3a,b*).

The pre-S1 elbow, embedded in the lipid bilayer, consists of two half transmembrane helices (half TM1 and half TM2). The half TM1 connects to LD9, which is the last alpha helix in the LD; the half TM2 connects to the pre-S1 helix, a short alpha helix prior to S1 running horizontally along the intracellular face of the membrane (*Figures 2a*, *3c and d*). This unique configuration pulls the intracellular half of S1 away from the pore center, resulting in a hydrophobic pocket behind the pre-S1 elbow and surrounded by half TM1 and S1 (*Figure 3c*). Moreover, the outward movement of S1 opens a

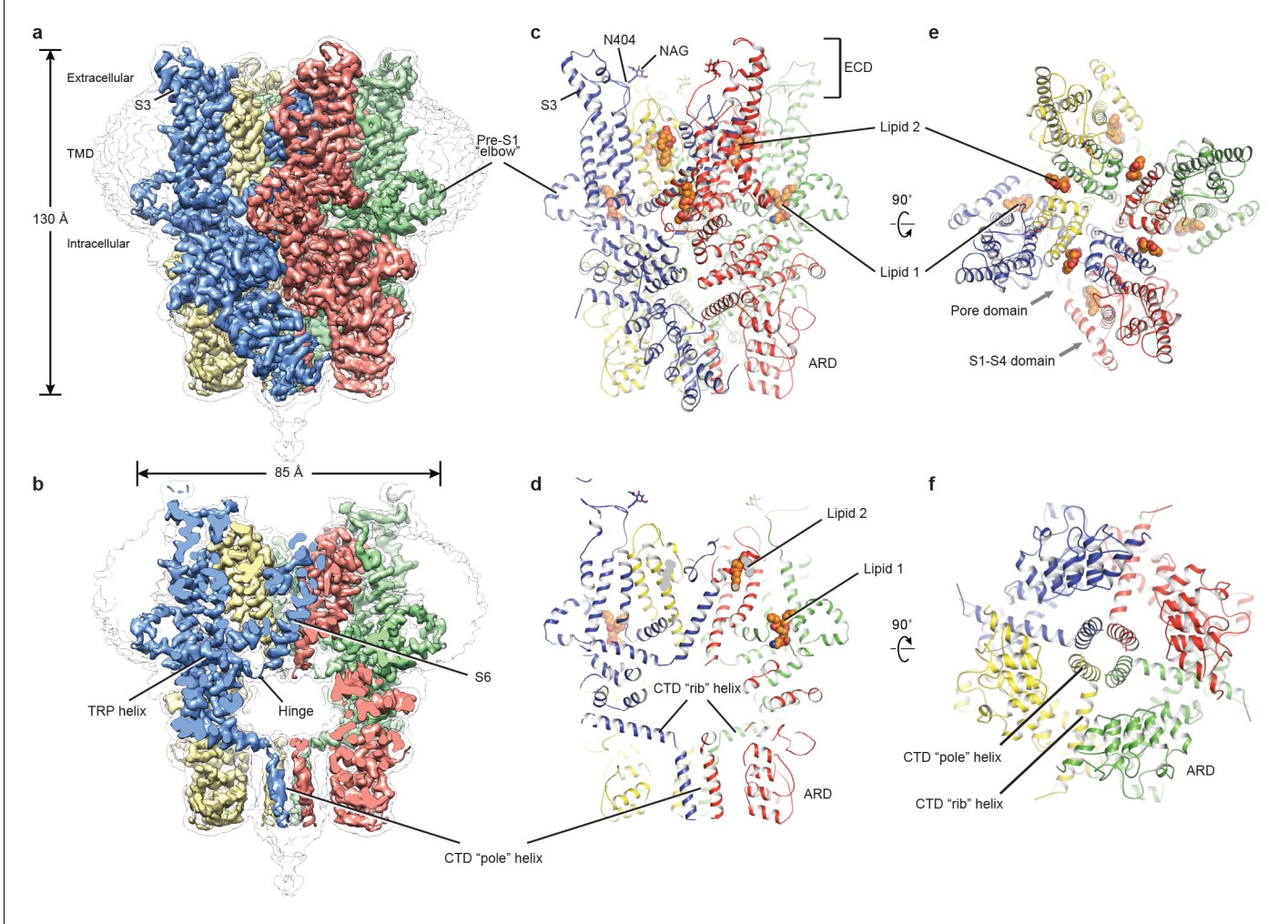

**Figure 1.** Architecture of human TRPC3. (**a**) Three-dimensional reconstruction viewed parallel to the membrane. The transparent envelope denotes the unsharpened reconstruction. (**b**) Slice view of the reconstruction showing the interior of the channel. (**c–f**) Atomic model of TRPC3 viewed parallel to the membrane (**c–d**), from the extracellular side (**e**), and from the intracellular side (**f**). Each subunit is colored differently.

DOI: https://doi.org/10.7554/eLife.36852.002

The following figure supplements are available for figure 1:

**Figure supplement 1.** Preparation, electrophysiological characterization of human full-length TRPC3.

DOI: https://doi.org/10.7554/eLife.36852.003

**Figure supplement 2.** Cryo-EM analysis of human full-length TRPC3.

DOI: https://doi.org/10.7554/eLife.36852.004

**Figure supplement 3.** Cryo-EM map of human full-length TRPC3.

DOI: https://doi.org/10.7554/eLife.36852.005

**Figure supplement 4.** Secondary structure arrangement of human TRPC3 and sequence alignment of TRPC family channels.

DOI: https://doi.org/10.7554/eLife.36852.006

window between itself and S5 from the adjacent subunit, exposing the intracellular half of S4 and the S4-S5 linker, which are key regions for TRP channel gating, to the lipid environment (*Figure 3d*).

Indeed, we observed a lipid-shaped density (lipid 1) in this pocket (*Figure 3c,d*). The head group of lipid 1 is well defined in the density map, forming several hydrogen bonds and polar interactions with residues in the LD9, the pre-S1 elbow, half TM1, and the S4-S5 linker, while the two hydrocarbon tails are in contact with S1, S4, the pre-S1 elbow, and half TM1 (*Figure 3e,f*). A similar pre-S1 elbow structure with lipid-like density has been observed in the *Drosophila* mechanosensitive channel NOMPC (*Jin et al., 2017*). We suggest that this lipid site may be crucially linked to channel

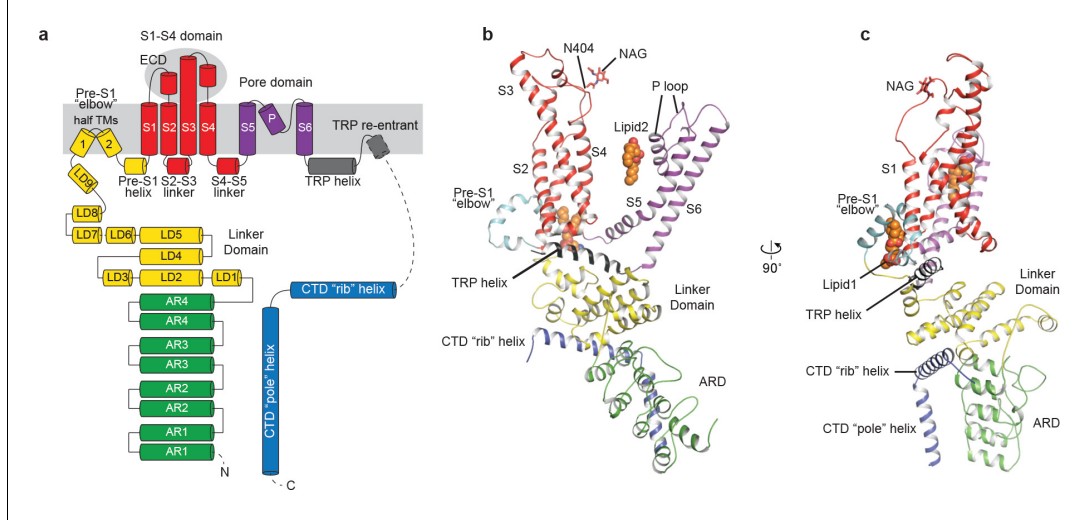

**Figure 2.** Structure of a single subunit. (a) The schematic representation of TRPC3 domain organization. Dashed lines indicate the regions that have not been modeled. (b–c) Cartoon representation of one subunit color-coded to match panel a.

DOI: https://doi.org/10.7554/eLife.36852.007

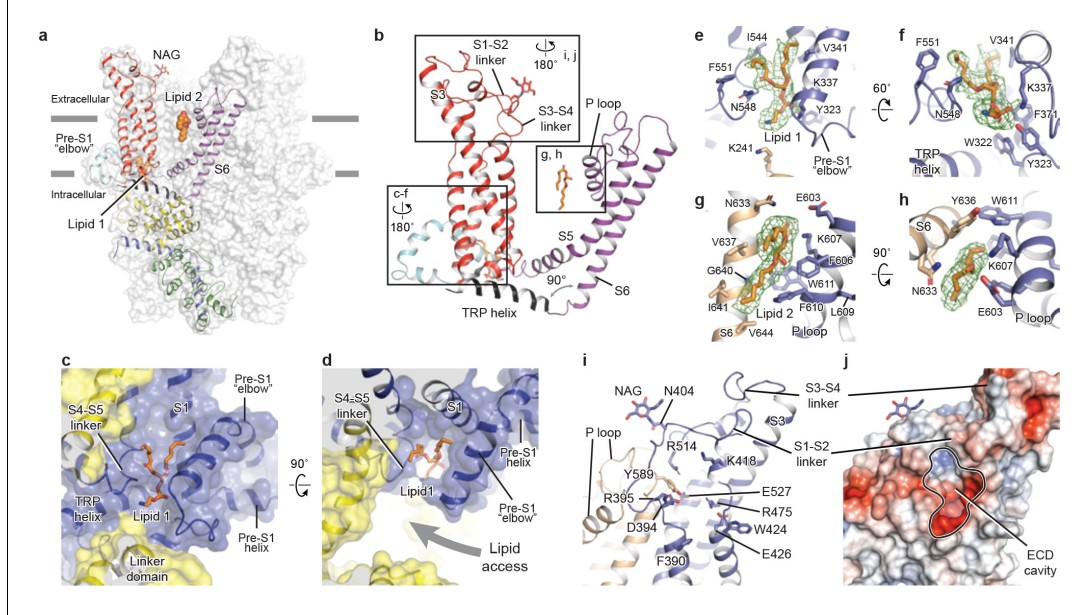

**Figure 3.** Transmembrane domain, extracellular domain, and lipid-binding sites. (a) Domain organization. The channel is shown in surface representation, with one subunit shown in cartoon representation. The colors match those in *Figure 2a*. (b) Details of the transmembrane domain and extracellular domain. (c–d) Pre-S1 elbow and binding site of lipid 1. The lipid molecule is buried inside the pocket formed by pre-S1 elbow, S1, and the S4-S5 linker. Two adjacent subunits (blue and yellow) are shown in both cartoon and surface representations. The lipid molecule is shown as sticks. (e–f) Residues that interact with lipid 1 are shown in sticks, and protein is shown in cartoon representation. Lipid density is shown in mesh. (g–h) Lipid 2 binds between S6 and the P loop of adjacent subunits, which are in light blue and wheat. (i) Structure of the ECD. Key residues forming the cavity is shown in sticks. Adjacent subunits are in light blue and wheat. (j) Surface representation of the ECD, colored according to the electrostatic surface potential. The color gradient is from −5 to 5 kT/*e* (red to blue).

DOI: https://doi.org/10.7554/eLife.36852.008

The following figure supplement is available for figure 3:

**Figure supplement 1.** Comparison of the TMD of TRPC3 with TRPV1.

DOI: https://doi.org/10.7554/eLife.36852.009

activation, given its interaction with S4 and the S4-S5 linker. A mutation in this region (T561A on S4) results in gain of function, causing abnormal Purkinje cell development and cerebellar ataxia in moonwalker mice (*Becker, 2014*) (*Figure 1—figure supplement 4*).

We also identified a second lipid-like density (lipid 2) in the lateral fenestration of the pore domain, wedged between the P loop and S6 of adjacent subunit and forming both hydrophobic and hydrophilic interactions (*Figure 3g,h*). Specifically, G640 on S6, interacting directly with the hydrocarbon tail of lipid 2, has been reported as a key determinant of lipid recognition in TRPC3. Mutations of G640 to alanine or larger residues distinctly changed the sensitivity of channel to lipid activators (*Lichtenegger et al., 2018*). Moreover, lipid 2 is in close contact with the LFW motif on the P loop, which is highly conserved throughout the TRPC family and is crucial to channel function (*Figure 1—figure supplement 4*). Replacing this motif by three alanine residues in TRPC5 and TRPC6 resulted in a nonfunctional channel (*Strübing et al., 2003*). Therefore, the lipid 2 binding site likely represents another important modulation site. In addition to interaction with lipid 2, the LFW motif forms multiple hydrophobic interactions within the pore domain and therefore plays an important role in maintaining the proper structure of the pore domain (*Figure 3i*).

A second unique feature of TRPC3 is the remarkably long S3, stretching out into the extracellular side and supporting the formation of the ECD (*Figure 3a*). Within the ECD we observed a cavity-like feature (*Figure 3j*), with S3 and the S3-S4 linker as a 'back wall and roof', and the S1-S2 linker forming the entrance. This cavity is located right above the lipid bilayer, and its interior is filled with both charged and hydrophobic residues (*Figure 3j*). Moreover, a tyrosine residue (Y589) in the loop connecting the S5 and the P loop plugs into the cavity (*Figure 3i*). We speculate that the cavity may serve as a binding site for small molecules and that binding of small molecules may directly affect channel function through Y589, implying a role for the ECD as a sensor of external stimuli. This is in line with the finding that Pyr3, a TRPC3-specific inhibitor, likely binds to the extracellular side of the protein (*Kiyonaka et al., 2009*). Furthermore, a glycosylation site (N404) is observed in the S1-S2 loop, consistent with the prediction that TRPC3 is monoglycosylated in the extracellular side (*Figure 3i*) (*Vannier et al., 1998*). The site is very close to the P loop, suggesting that the glycosylation status may affect channel activity, and this is consistent with the report that N-linked glycosylation is a key determinant of the basal activity of TRPC3 (*Dietrich et al., 2003*). Further studies are necessary to clarify the physiological role of the ECD.

## TRP domain

The TRP domain—the namesake region in the TRP channel located at the border between the transmembrane domain and the intracellular domain—is crucially involved in signal transduction and channel gating (*García-Sanz et al., 2007*; *Taberner et al., 2013*). Similar to that of TRPM4, the TRP domain consists of a TRP helix that runs nearly parallel along the intracellular face of the membrane and a TRP re-entrant helix embedded in the lipid bilayer (*Autzen et al., 2018*; *Guo et al., 2017*; *Winkler et al., 2017*) (*Figure 4a*). The TRP helix penetrates into the tunnel formed by the S4-S5 linker of the TMD on the top and the LD9 of the linker domain in the intracellular space on the bottom (*Figure 4a*), showing an apparently disengaged connection to the S6 helix through a loop of the hinge region instead of a continuous alpha helical structure as in TRPM4 (*Figure 4b,e*). While the densities for both S6 and TRP helix were well defined, their linker region was surprisingly poorly defined, indicating a high flexibility between the TRP helix and S6 (*Figures 4b* and *1b*). The TRP helix forms an approximate right angle to the S6, in strong contrast to the TRPV1, TRPA1, and TRPM4 structures whose TRP helices form obtuse angles with S6 (*Figure 4b–e*). Such a unique configuration of TRP helix and S6 in TRPC3 has two consequences. First, the upward tilting of the TRP helix allows itself approaching to the S4-S5 linker, suggesting their tighter coupling in comparison to the other TRP subfamily channels (*Figure 4f–h*). Second, the C-terminus of the TRP helix is in close contact with the lipid 1 site. Given the crucial role of TRP helix and S4-S5 linker in channel gating and their possible involvement in voltage dependence (*Nilius et al., 2005b*), the interplay among the TRP helix, the S4-S5 linker and the lipid 1 site may provide a molecular basis for the lipid-sensitive gating mechanism of TRPC3 relative to other TRP subfamily channels (*Itsuki et al., 2012*).

Furthermore, the TRP helix forms a series of polar and hydrophobic interactions with the S4-S5 linker and the LD9 helix (*Figure 4a*). Specifically, the highly conserved tryptophan W673 is extensively coupled with the S4-S5 linker through interactions with G552, P553, and P546. Mutation of the corresponding W673 in TRPV3 results in Olmsted syndrome (*Ni et al., 2016*), and replacement of

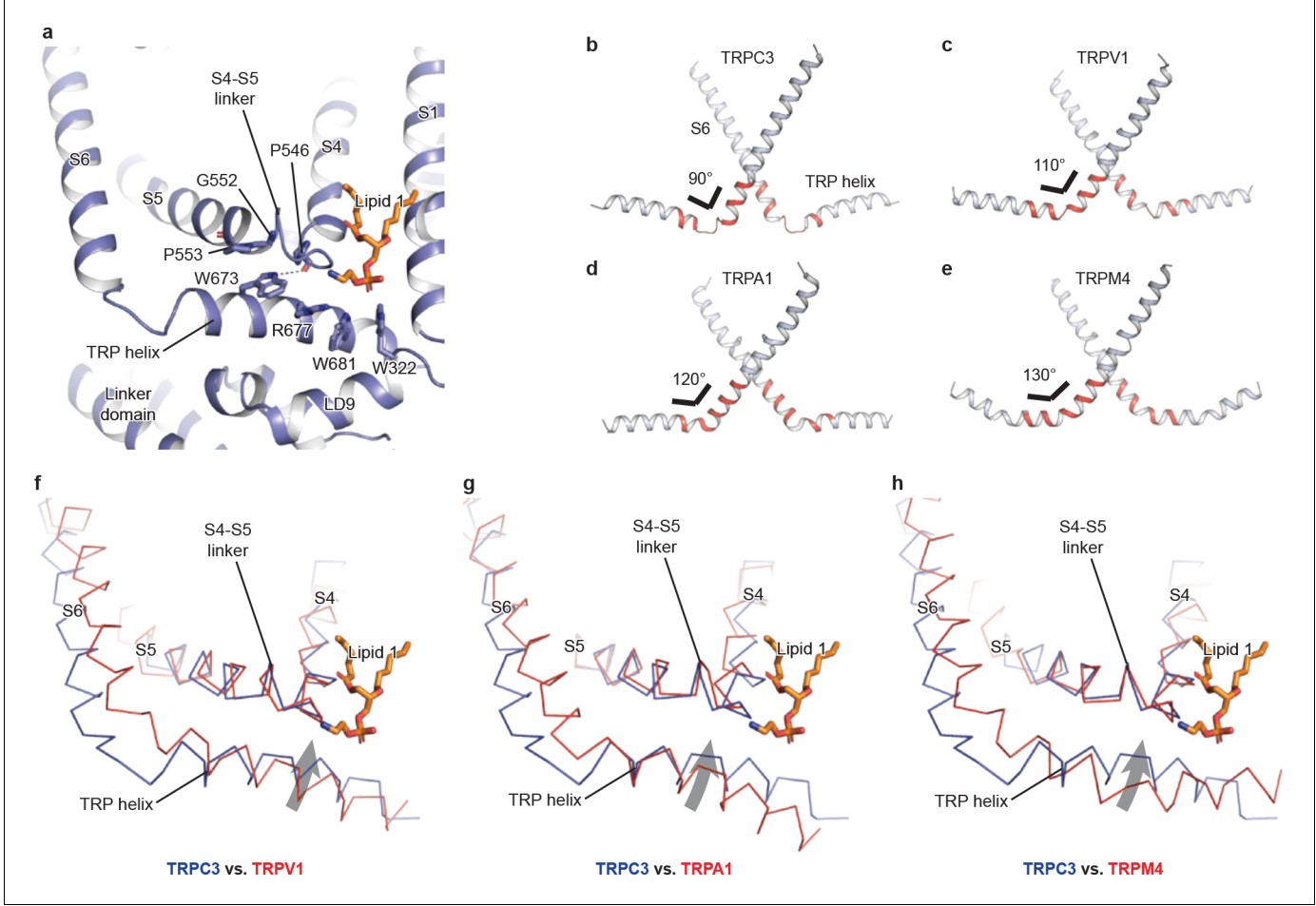

**Figure 4.** The TRP domain. (**a**) Cartoon representation of the TRP helix, pre-S1 elbow, TMD, and linker domain, showing their interaction. Lipid 1 is shown in sticks. W673 on the TRP helix stacks with P553 and G552 forming a hydrogen bond with the backbone oxygen (dashed line) of P546 on the S4-S5 linker. The side chain of R677 is in close contact with the head group of lipid 1. (**b–e**) The pore lining helix S6 and the TRP helix in TRPC3 (**b**), TRPV1 (**c**), TRPA1 (**d**), and TRPM4 (**e**). The angle between the S6 and TRP helices are indicated; only two subunits are shown for clarity. The hinge connecting the S6 and TRP helix is highlighted in red. (**f–h**) Comparison of TRPC3 with TRPV1, TRPA1, and TRPM4, respectively, focusing on the S4, S5 and TRP helix. Structures are superimposed using backbone atoms in S4 and S5. TRPC3 is in blue, whereas TRPV1, TRPA1 and TRPM4 are in red. Proteins are shown in ribbon representation, and lipid 1 in TRPC3 is shown in sticks. Arrows indicate the relative movement of the TRP helix in TRPC3 compared to TRPV1, TRPA1 or TRPM4.

DOI: https://doi.org/10.7554/eLife.36852.010

the corresponding residue in NOMPC results in a channel that has increased current amplitude but is nonresponsive to mechanical stimuli (*Jin et al., 2017*). Mutation of the corresponding tryptophan in TRPV1 abolishes channel activation in response to depolarization (*Gregorio-Teruel et al., 2014*). Replacement of the corresponding G552 in TRPC4 and TRPC5 by serine results in a constantly open channel (*Beck et al., 2013*). Another highly conserved tryptophan residue in the TRP helix, W681, tightly packs with W322 in the LD9 (*Figure 4a* and *Figure 1—figure supplement 4*). Interestingly, the highly conserved R677 in the TRP helix is close to the head group of lipid 1, and its replacement by histidine increases channel activity and results in neuronal cell death and cerebellar ataxia, perhaps by affecting the binding of lipid 1 (*Figure 4a*) (*Fogel et al., 2015*).

## Ion-conducting pore

The ion-conducting pore of TRPC3 is lined with an extracellular selectivity filter and an intracellular gate, with a wide central vestibule in the middle (*Figure 5a*). The pore adopts a closed conformation with the narrowest radius - at I658 and L654 on S6 close to the intracellular exit − of less than 1 Å,

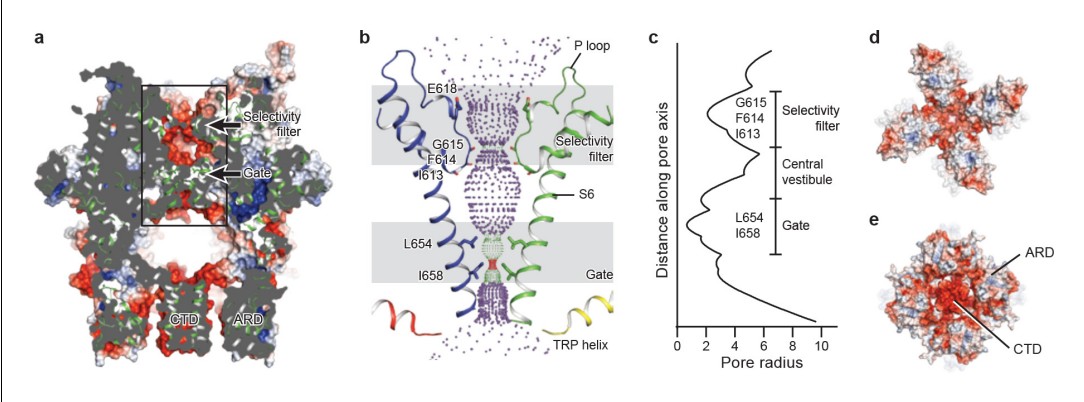

**Figure 5.** The ion-conducting pore. (a, d, e) Surface representation of TRPC3, viewed (a) parallel to the membrane, (d) from the extracellular side, and (e) from the intracellular side. The surface is colored according to electrostatic surface potential; the color gradient is from −5 to 5 kT/$e$ (red to blue). The protein is also shown in cartoon representation in (a). (b) The shape and size of the ion-conducting pore (boxed area in panel a). The P loop and S6 of two subunits and the TRP helix of the other two subunits are shown as cartoons, and the side chains of restriction residues are shown as sticks. Purple, green, and red spheres define radii of >2.3, 1.2–2.3, and <1.2 Å, respectively. (c) Plot of pore radius as a function of distance along the pore axis in Angstroms.

DOI: https://doi.org/10.7554/eLife.36852.011

thus preventing ion passage (*Lichtenegger et al., 2013*) (*Figure 5b*). Presumably, the channel is trapped in a lipid-bound inactive state or the bound lipids are not the activator DAG. The selectivity filter is defined by the backbone carbonyl oxygens of I613, F614 and G615 located in the P loop. The narrowest point at G615 has a radius of 2.1 Å, allowing partially dehydrated ions to pass through (*Figure 5b,c*). Moreover, five acidic residues in the P loop and the extracellular end of S6 in TRPC3 impose a negative electrostatic surface potential, which is important for cation selectivity (*Figure 5a,d*). On the intracellular site, the inner surface along the CTD and ARD contains acidic residues, giving rise to a negative charge and thus providing a possible pathway by which cations can access the cytoplasm (*Figure 5a,e*).

Similar to other $Ca^{2+}$-permeable TRP channels, an acidic residue, E618, is located at the entrance of the selectivity filter. An E618Q mutation impedes the calcium permeability of TRPC3, but it preserves monovalent permeation (*Poteser et al., 2011*) (*Figure 5b*). The neutralization of the corresponding acidic amino acid on TRPV1 remarkably decreases channel's permeability to divalent ions (*García-Martínez et al., 2000*). By contrast, replacement of a glutamine residue at the corresponding position (Q977) by an acidic amino acid in TRPM4, which is a $Ca^{2+}$-impermeable TRP channel, produced moderate $Ca^{2+}$ permeability (*Nilius et al., 2005a*). Thus, having an acidic residue close to the selectivity filter may represent a general principle of permeability for divalent cations in nonselective $Ca^{2+}$-permeable TRP channels.

## The intracellular domain

TRPC3 exhibits a similar intracellular domain composition as TRPA1, including a C-terminal CTD and a N-terminal ARD. We found several unanticipated features that advance our understanding of the molecular basis of TRPC family (*Figure 6a*). First, the ARD in TRPC3, consisting of 4 ARs, is significantly shorter than that in TRPA1 (16 repeats). Second, instead of a straight coiled-coil domain as in TRPA1, TRPC3 adopts the characteristic umbrella-like CTD 'pole' and 'rib' domain of the TRPM family (*Figure 6b,c*). Interestingly, the turn from the pole to the rib helix is where the ankyrin repeats end. Third, between the rib domain and TRP helix, there is a linker domain that has remarkable structural similarities to the MHR4 (TRPM homology region) domain in TRPM4 (*Autzen et al., 2018*; *Guo et al., 2017*; *Winkler et al., 2017*), as well as to the linker domain in NOMPC (*Jin et al., 2017*). The location of the linker domain suggests a role for signal transduction from ARD and CTD further to TMD. Overall, the TRPC3 forms a unique intracellular domain that has structural features characteristic of the TRPM, TRPA, and NOMPC families. Although the functional role of the intracellular domain is yet unknown, it clearly contributes to the channel assembly through three major interfaces.

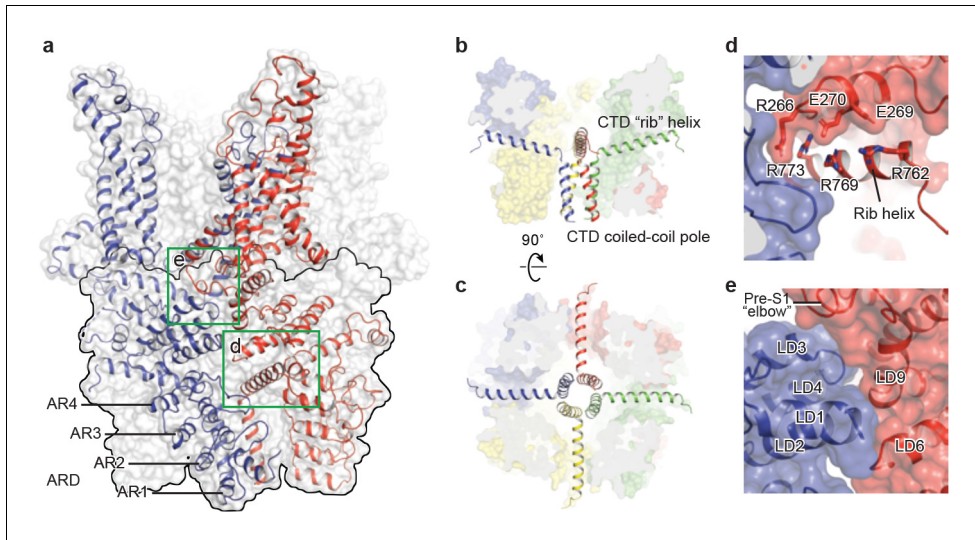

**Figure 6.** The intracellular domain. (**a**) Surface representation of TRPC3 with two adjacent subunits shown in cartoon representation. The intracellular domain is highlighted in the black frame. Two interfaces highlighted in green frames are enlarged in (**d–e**). (**b–c**) Cartoon representation of the CTD coiled-coil pole and the rib helix. The intracellular domain is shown in surface representation, viewed in parallel to the membrane (**b**) and from the intracellular side (**c**). (**d**) Inter-subunit interface formed by the CTD rib helix with adjacent ARD and LD. Protein is shown in cartoon and surface representations. Two adjacent subunits are in blue and red. Charged residues forming hydrogen bond or polar interaction with each other are shown as sticks. (**e**) Interface between adjacent LDs and pre-S1. Alpha helices involved in the inter-subunit interaction are indicated.

DOI: https://doi.org/10.7554/eLife.36852.012

The first interface is contributed by the vertical CTD pole helices of the four subunits winding into a tetrameric coiled-coil assembly (*Figure 6b,c*). This is a common feature employed to specify subunit assembly and assembly specificity within the voltage-gated ion channel superfamily (*Figure 6b,c*). The second interface is formed by the horizontal CTD rib helix penetrating through the tunnel composed of ARD and LD from neighboring subunits, thus tethering them together (*Figure 6c,d*). Notably, the rib helix is rich in positively charged residues, forming multiple interactions with the charged residues in the LD. The third interface is located between LD and LD/pre-S1 elbow of the adjacent subunit (*Figure 6e*). All these interactions knit the tetramer together.

## Discussion

The TRPC3 structure displays several unique features. Distinct to the TRPM, TRPV or TRPA channels whose TRP helix and S6 form a continuous alpha helical structure, the TRP helix in TRPC3 is disengaged from the S6, and is in close contact to both the S4-S5 linker and lipid 1 site, which perhaps links to the lipid-induced activation. The remarkably long S3 endows TRPC3 a unique shape of TMD and frames the ECD in which a cavity may act as a binding site for small molecules, suggesting a role for the ECD in sensing extracellular stimuli. We identified two lipid binding sites, one buried in a pocket surrounded by the pre-S1 elbow, S1, and the S4-S5 linker, and the other inserted into the lateral fenestration of the pore domain. Our structure provides a framework for understanding the complex gating mechanism of TRPC3.

## Materials and methods

### Construct, expression and purification of TRPC3

A full-length human *TRPC3* gene (UniProtKB (http://www.uniprot.org) accession number, Q13507) with 836 amino acid residues was synthesized by Genscript and was subcloned into a modified

version of pEG BacMam vector containing: a twin strep-tag, a His8-tag, and green fluorescent protein (GFP) with thrombin cleavage site at the N terminus (*Goehring et al., 2014*). The recombinant Bacmid DNA and baculovirus of TPRC3 were generated by sf9 insect cells, and P2 viruses were used to infect suspension HEK293 cells. The recombinant Bacmid DNA and baculovirus of TPRC3 were generated by sf9 insect cells, and P2 viruses were used to infect suspension HEK293 cells. The HEK293 and sf9 were obtained from the ATCC and were not re-authenticated or tested for mycoplasma after purchase.

For large-scale expression, suspension HEK293 cells were cultured in Freestyle 293 expression Medium (Invitrogen) with 1% (v/v) fetal bovine serum (FBS). When cell density reached around 3 million/ml, 8% (v/v) of P2 viruses were introduced. At 12 hr post-infection, 10 mM sodium butyrate was supplemented and then cells were transferred to 30°C. The infected cells were collected at 48 hr post-infection by centrifugation at 4000 rpm for 15 min at 4°C and then were washed once with TBS buffer (20 mM Tris, pH 8.0, 150 mM NaCl).

TRPC3 was extracted from the cells by solubilization buffer containing 20 mM Tris 8.0, 500 mM NaCl in the presence of 1 mM PMSF, 0.8 μM aprotinin, 2 μg/mL leupeptin, and 2 mM pepstatin A with 1% digitonin (Calbiochem) for 2 hr at 4°C. The cell debris were eradicated by ultracentrifugation at 40,000 rpm using 45 Ti rotor (Beckman Coulter, Inc) for 1 hr at 4°C. The solubilized proteins were incubated with TALON resin and the resin was washed with 10 column volumes of wash buffer (20 mM Tris 8.0, 500 mM NaCl, 15 mM imidazole, and 0.1% digitonin). The TALON resin-bound TRPC3 was eluted with elution buffer (20 mM Tris 8.0, 500 mM NaCl, 250 mM imidazole, and 0.1% digitonin). Thrombin (1:20 molar ratio) and 10 mM EDTA were added into the eluted sample and incubated for 3 hr on the ice. In order to further purify the protein, the sample was concentrated and loaded onto a superpose6 column in buffer containing 20 mM Tris 8.0, 500 mM NaCl, 1 mM EDTA with 0.1% digitonin. Peak fractions containing TRPC3 were pooled and concentrated to 5 mg/mL.

## EM sample preparation and data acquisition

The purified TRPC3 protein sample (2.5 μL) at a concentration of 5 mg/mL was applied onto a glow-discharged Quantifoil holey carbon grid (gold, 1.2/1.3 μm size/hole space, 300 mesh). The gird was blotted for 1.5 s at 100% humidity by using a Vitrobot Mark III, and then was plunged into liquid ethane cooled by liquid nitrogen. Images were obtained by an FEI Titan Krios electron microscope operating at 300 kV with a nominal magnification of 130,000 × Gatan K2 Summit direct electron detector was used in order to record image stacks in super-resolution counting mode with a binned pixel size of 1.074 Å. Every image was dose-fractionated to 40 frames with a total exposure time of 8 s with 0.2 s per frame. Dose rate was 6.76 e$^-$ Å$^{-2}$ s$^{-1}$. The images stacks were recorded using the automated acquisition program SerialEM (*Mastronarde, 2005*). Nominal defocus values varied from 1.0 to 2.5 μm.

## EM data processing

MortionCor2 was used to implement motion-correction of summed movie stacks (*Zheng et al., 2017*). Gctf was applied to estimate Defocuse values (*Zhang, 2016*). Particles were picked from approximately 200 micrographs using Gautomatch (http://www.mrc-lmb.cam.ac.uk/kzhang/Gautomatch/) and subjected to an initial reference-free 2D classification using Relion 2.1 (*Scheres, 2012*). Nine representative 2D class averages were selected as templates for automated particle picking for the entire data set using Gautomatch. The auto-picked particles were visually checked and obvious bad particles were manually removed. The picked particles were cleaned up throughout three rounds of 2D classification. CryoSPARC was applied to obtain an initial model (*Punjani et al., 2017*). The selected particles after 2D classification were subjected to 3D classification of five classes using Relion 2.1, with the initial reconstruction low-pass-filtered to 60 Å as a reference model. Only one out of five classes presented high-resolution features, hence, particles from this class were combined and further refined via Relion 2.1. Particles were further refined using the local refinement from Frealign with C4 symmetry applied and high-resolution limit for particle alignment set to 4.5 Å (*Grigorieff, 2016*). The resolutions reported are based on the 'limiting resolution' procedure in which the resolution during refinement is limited to a lower resolution than the resolution estimated for the final reconstruction. The final resolutions reported in *Table 1* are based on the gold standard Fourier shell correlation (FSC) 0.143 criteria. To calculate the FSC plot, a soft mask (4.3 Å extended

**Table 1.** Statistics of EM data processing and model refinement.

| Data collection/processing | |
| --- | --- |
| Microscope | Titan Krios (FEI) |
| Voltage (kV) | 300 |
| Defocus range (µM) | 1.0–2.5 |
| Exposure time (s) | 8 |
| Dose rate (e⁻/Å²/s) | 6.76 |
| Number of frames | 40 |
| Pixel size (Å) | 1.074 |
| Particles refined | 143855 |
| Resolution (Å) | 3.3 |
| FSC threshold | 0.143 |
| Resolution range (Å) | 412.4–3.3 |
| Model statistics | |
| Number of atoms | 20988 |
| Protein | 20744 |
| Ligand | 244 |
| r.m.s. deviations | |
| Bond length (Å) | 0.005 |
| Bond angle (°) | 1.008 |
| Ramachandran plot | |
| Favored (%) | 94.09 |
| Allowed (%) | 5.77 |
| Disallowed (%) | 0.14 |
| Rotamer outlier (%) | 0.85 |
| Clashscore | 3.0 |

DOI: https://doi.org/10.7554/eLife.36852.013

from the reconstruction with an additional 4.3 Å cosine soft edge, low-pass filtered to 10 Å) was applied to the two half maps.

## Model building

The model of TRPC3 was built in Coot using the TMD domain of TRPM4 structure (PDB 5wp6) as a guide (*Emsley et al., 2010*). De novo building was mainly guided by bulky residues and secondary structure prediction (*Figure 1—figure supplement 4*). The TRPC3 structure chiefly consists of α helices, which greatly assisted register assignment. In the initial de novo-built model, the order and length of the secondary structure features, as well as the positions of bulky residues within each secondary structure feature are in good agreement with the prediction (*Figure 1—figure supplement 4*). The initial model was then subjected to real space refinement using Phenix.real_space_refine with secondary structure restraints (*Afonine et al., 2012*). The refined model was manually examined and re-modified via COOT. For validation of refined structure, FSC curves were applied to calculate the difference between the final model and EM map. The geometries of the atomic models were evaluated using MolProbity (*Chen et al., 2010*). All figures were prepared using UCSF Chimera and Pymol (Schrödinger) (The PyMOL Molecular Graphics System) (*Pettersen et al., 2004*).

## Electrophysiology

Suspension HEK293 cells were cultured in Freestyle 293 expression Medium (Invitrogen, Waltham, Massachusetts, USA) with 1% (v/v) fetal bovine serum (FBS). When cell density reached around 1

million/ml, 5% (v/v) of P2 viruses of human TRPC3 were introduced. Infected cells were incubated in 24-well plate at 37°C, and were recorded 12–24 hr post-infection. Whole-cell voltage clamp recordings were performed using a HEKA EPC-10 amplifier at room temperature. The holding potential was +60 mV. The electrodes were filled with internal solution containing (mM) 130 CsOH, 130 glutamate, 3.1 $MgCl_2$, 2.8 $CaCl_2$, 10 EGTA, 2 $ATPNa_2$, 0.3 $GTPNa_2$, 10 HEPES (pH 7.2 adjusted with CsOH). The bath solution contains (mM) 140 NaCl, 5 KCl, 2 $CaCl_2$, 1 $MgCl_2$, 10 glucose, 10 HEPES (pH 7.4 adjusted with NaOH). Solution change was done using a two-barrel theta-glass pipette controlled manually. Data were acquired at 10 kHz using Patchmaster software (HEKA). Data were filtered at 1 kHz, and analyzed with Axograph software (www.axograph.com).

## Data availability

The cryo-EM density map and coordinate of TRPC3 have been deposited in the Electron Microscopy Data Bank (EMDB) accession number EMD-7620, and in the RCSB Protein Data Bank (PDB) accession code 6CUD.

# Acknowledgements

We thank G Zhao and X Meng for the support with data collection at the David Van Andel Advanced Cryo-Electron Microscopy Suite. We appreciate the VARI High-Performance Computing team for computational support. We thank D Nadziejka for technical editing, and C-H Lee for helpful discussion. This work was supported by internal VARI funding.

# Additional information

## Funding

| Funder | Author |
| --- | --- |
| Van Andel Research Institute | Wei Lü |

The funders had no role in study design, data collection and interpretation, or the decision to submit the work for publication.

## Author contributions

Chen Fan, Wooyoung Choi, Formal analysis, Writing—review and editing; Weinan Sun, Formal analysis; Juan Du, Wei Lü, Conceptualization, Resources, Data curation, Software, Formal analysis, Supervision, Funding acquisition, Validation, Investigation, Visualization, Methodology, Writing—original draft, Project administration, Writing—review and editing

## Author ORCIDs

Wei Lü https://orcid.org/0000-0002-3009-1025

## Decision letter and Author response

Decision letter https://doi.org/10.7554/eLife.36852.020
Author response https://doi.org/10.7554/eLife.36852.021

# Additional files

## Supplementary files

• Transparent reporting form
DOI: https://doi.org/10.7554/eLife.36852.014

## Data availability

The cryo-EM density map and coordinate of TRPC3 have been deposited in the Electron Microscopy Data Bank (EMDB) accession number EMD-7620, and in the RCSB Protein Data Bank (PDB) accession code 6CUD.

The following datasets were generated:

| Author(s) | Year | Dataset title | Dataset URL | Database and Identifier |
|---|---|---|---|---|
| Lü W | 2018 | Cryo-EM density map and coordinate of TRPC3 | http://www.ebi.ac.uk/pdbe/entry/emdb/EMD-7620 | Electron Microscopy Data Bank, EMD-7620 |
| Lü W | 2018 | Cryo-EM density map and coordinate of TRPC3 | http://www.rcsb.org/pdb/search/structid-Search.do?structureId=6CUD | RCSB Protein Data Bank, 6CUD |

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
