## [Decision Letter]

Thank you for submitting your article "Structure of the human lipid-sensitive cation channel TRPC3" for consideration by *eLife*. Your article has been favorably evaluated by Richard Aldrich (Senior Editor) and three reviewers, one of whom, Leon D. Islas (Reviewer #1), is a member of our Board of Reviewing Editors.

The reviewers have discussed the reviews with one another and the Reviewing Editor has drafted this decision to help you prepare a revised submission.

Summary:

TRPC channels are non-selective Ca^2+^-permeable channels widely expressed in the body, and participate in many important physiological functions, including a role in store-operated Ca^2+^-entry. The closest homologues in the TRPC family, TRPC3, C6 and C7, are activated by diacylglycerol (DAG) generated from the degradation of PIP2, and the lipid binding sites remain unknown. There is currently limited information on the mechanisms of gating of these channels and on their pharmacology, and no high-resolution structural information is available. In the present study, Chen Fan and co-workers have determined the structure of the full-length human TRPC3 channel in a closed conformation at an overall resolution of 3.3A using single-particle cryo-EM. Most of the channel residues were well resolved to allow for model building, exhibiting an overall architecture that is similar in its transmembrane domain to that of other TRP channels, with a domain-swapped arrangement between the S1-S4 helices and the pore domain.

The manuscript thus offers a set of novel observations that could direct further understanding of the mechanisms of function of this group of TRP channels.

Essential revisions:

1) No biochemical data on the purity and quality of the sample is provided. The authors should show a coomassie-stained gel to show the purity of the sample and provide size-exclusion profiles for their prep.

2) No functional data is provided in the manuscript. Although the reported structure is for a full-length WT channel, the authors should show that the baculovirus-expressed channel is functional in HEK 293 cells.

3) More insights would be obtained from a more meaningful comparison to other available TRP channel structures, especially regarding the conclusion that S6-TRP helix form a 90 degree angle in TRPC3 only.

---

## [Author Response]

Essential revisions:1) No biochemical data on the purity and quality of the sample is provided. The authors should show a coomassie-stained gel to show the purity of the sample and provide size-exclusion profiles for their prep.

We have now included a new supplemental figure (Figure 1—figure supplement 1A, B), showing the size-exclusion profile of hTRPC3 and a coomassie-stained SDS gel of purified hTRPC3.

2) No functional data is provided in the manuscript. Although the reported structure is for a full-length WT channel, the authors should show that the baculovirus-expressed channel is functional in HEK 293 cells.

Thanks for the suggestion. We have performed electrophysiology experiments using baculovirus-infected HEK293 cells, and we were able to record OAG-invoked currents that are blocked by TRPC3-specific inhibitor Pyr3 (Figure 1—figure supplement 1C). Our result shows that the baculovirus-expressed hTRPC3 is functional in HEK293 cells.

3) More insights would be obtained from a more meaningful comparison to other available TRP channel structures, especially regarding the conclusion that S6-TRP helix form a 90 degree angle in TRPC3 only.

We thank the reviewers for this comment. We have now made a more careful comparison of the S6-TRP helix with other available TRP channel structures (Figure 4 F-H) and added the following to the Discussion:

“Such a unique configuration of TRP helix and S6 in TRPC3 has two consequences. […] Given the crucial role of TRP helix and S4-S5 linker in channel gating and their possible involvement in voltage dependence, the interplay among the TRP helix, the S4-S5 linker and the lipid 1 site may provide a molecular basis for the lipid-sensitive gating mechanism of TRPC3 relative to other TRP subfamily channels.”